# Impact of Climate Change on Schistosomiasis Transmission and Distribution—Scoping Review

**DOI:** 10.3390/ijerph22050812

**Published:** 2025-05-21

**Authors:** Kwame Kumi Asare, Muhi-Deen Wonwana Mohammed, Yussif Owusu Aboagye, Kathrin Arndts, Manuel Ritter

**Affiliations:** 1Biomedical and Clinical Research Centre, College of Allied Health Sciences, University of Cape Coast, Cape Coast, Ghana; muhideenm7@gmail.com (M.-D.W.M.); aboagye.yussif13@gmail.com (Y.O.A.); 2Department of Immunology, Noguchi Memorial Institute for Medical Research, University of Ghana, Accra, Ghana; 3Institute for Medical Microbiology, Immunology and Parasitology (IMMIP), University Hospital Bonn (UKB), 53127 Bonn, Germany; kathrin.arndts@ukbonn.de; 4German-West African Centre for Global Health and Pandemic Prevention (G-WAC), Partner site Bonn, 53127 Bonn, Germany

**Keywords:** schistosomiasis, climate change, transmission dynamics, snail population, emerging hotspots, climate adaptation policies, neglected tropical diseases (NTDs)

## Abstract

Schistosomiasis, a neglected tropical disease caused by parasitic worms of the genus *Schistosoma* and transmitted through freshwater snails, affects over 200 million people worldwide. Climate change, through rising temperatures, altered rainfall patterns, and extreme weather events, is influencing the distribution and transmission dynamics of schistosomiasis. This scoping review examines the impact of climate change on schistosomiasis transmission and its implications for disease control. This review aims to synthesize current knowledge on the influence of climate variables (temperature, rainfall, water bodies) on snail populations, transmission dynamics, and the shifting geographic range of schistosomiasis. It also explores the potential effects of climate adaptation policies on disease control. The review follows the Arksey and O’Malley framework and PRISMA-ScR guidelines, including studies published from 2000 to 2024. Eligible studies were selected based on empirical data on climate change, schistosomiasis transmission, and snail dynamics. A two-stage study selection process was followed: title/abstract screening and full-text review. Data were extracted on environmental factors, snail population dynamics, transmission patterns, and climate adaptation strategies. Climate change is expected to increase schistosomiasis transmission in endemic regions like Sub-Saharan Africa, Southeast Asia, and South America, while some areas, such as parts of West Africa, may see reduced risk. Emerging hotspots were identified in regions not currently endemic. Climate adaptation policies, such as improved water management and early warning systems, were found effective in reducing transmission. Integrating climate adaptation strategies into schistosomiasis control programs is critical to mitigating the disease’s spread, particularly in emerging hotspots and shifting endemic areas.

## 1. Introduction

Schistosomiasis is a neglected tropical disease that affects over 200 million people globally, predominantly in sub-Saharan Africa, Asia, and parts of South America [1,2]. It is caused by parasitic worms of the genus *Schistosoma*, which have a complex lifecycle involving vertebrate hosts and freshwater snails as intermediate hosts [3]. The disease results in significant morbidity, leading to chronic symptoms like anemia, malnutrition, organ damage, and, in severe cases, death [4]. Although major progress has been made in reducing its burden, schistosomiasis remains a persistent health challenge due to environmental, economic, and social factors that complicate disease control [1,5]. Among these factors, climate change stands out as a critical, often underestimated driver of schistosomiasis transmission dynamics, with the potential to influence disease distribution on a global scale [6,7].

Climate change, driven by increasing atmospheric greenhouse gas concentrations, is causing a rise in global temperatures, alterations in rainfall patterns, and an increased frequency of extreme weather events [8,9]. These environmental changes directly and indirectly affect ecosystems, wildlife, and human health [10]. For schistosomiasis, changing rainfall patterns, temperature fluctuations, and altered water availability can substantially influence freshwater snail hosts’ survival, reproduction, and distribution, which are central to the transmission cycle [11,12]. Higher temperatures and changes in precipitation can expand or shift the range of these snail populations, potentially leading to new schistosomiasis-endemic areas or intensifying transmission in existing ones [13]. Additionally, human-driven responses to climate change, such as the creation of dams, irrigation schemes, and other water management practices, further complicate the relationship between environmental changes and schistosomiasis dynamics [14].

Understanding how climate change affects the distribution and transmission of schistosomiasis is increasingly becoming important for public health, particularly in regions where the disease is endemic and resources for disease control are limited. Shifts in the geographic distribution of schistosomiasis due to climate-induced changes could result in the emergence of new transmission hotspots, potentially affecting previously unaffected populations [7,15]. Moreover, existing transmission zones may experience increased infection rates due to more favorable conditions for snail hosts, prolonging transmission seasons and potentially increasing disease prevalence [16,17]. Anticipating these changes requires an in-depth examination of the relationship between climate variables such as temperature, precipitation, and water availability and schistosomiasis transmission dynamics. Such insights can help in mapping current endemic regions and predicting future high-risk areas under various climate scenarios.

In recent years, several studies have examined the link between climate change and schistosomiasis [18,19,20,21]. These studies have highlighted how increasing temperatures and altered precipitation patterns can impact freshwater ecosystems and influence snail breeding sites, thereby affecting the spread and persistence of schistosomiasis [18,19,20]. For example, warmer temperatures may accelerate the lifecycle of snails and parasites, leading to faster transmission cycles and potentially higher rates of infection [22]. Increased rainfall and flooding may create additional habitats for snail populations, while drought and water scarcity can either reduce transmission by limiting snail habitats or concentrate human activities around fewer water sources, potentially increasing human-snail contact [23,24]. Despite these findings, there remains a significant gap in translating such knowledge into predictive models that can inform schistosomiasis control efforts. Many current studies focus on isolated aspects of climate change, and there is a lack of comprehensive reviews that bring together various factors to provide a holistic understanding of how climate change may alter schistosomiasis transmission on a regional or global scale.

Another dimension of the climate–schistosomiasis nexus is the role of climate adaptation and mitigation policies. Policies aimed at addressing climate change, such as water resource management, agricultural irrigation, and habitat conservation, can have unintended consequences for schistosomiasis transmission [13]. For example, the construction of dams and reservoirs to store water for dry seasons or to generate hydroelectric power may inadvertently create ideal habitats for snail populations, facilitating the spread of schistosomiasis [25]. Additionally, climate adaptation strategies that promote agricultural practices like expanded irrigation can lead to increased human–snail contact if such practices are implemented without considering their impact on local ecosystems [26]. As governments and organizations intensify climate adaptation and resilience efforts, it is essential to evaluate the potential impacts of these policies on schistosomiasis transmission and develop strategies to mitigate any adverse effects. Examining how climate adaptation efforts intersect with schistosomiasis control can help identify policies that safeguard public health while addressing environmental challenges.

Given the complex and evolving nature of these interactions, a scoping review is warranted to assess the current knowledge on the impact of climate change on schistosomiasis transmission and distribution. This review aims to systematically map existing evidence on how climate variables such as rainfall, temperature, and water availability influence schistosomiasis dynamics, with a focus on identifying key trends, gaps, and potential areas for further research. Additionally, the review will explore shifts in endemic areas due to climate-induced changes and examine projections of future transmission hotspots. Finally, this review will address the implications of climate adaptation policies on schistosomiasis control efforts, aiming to inform policymakers and public health officials about potential unintended consequences and highlight strategies for integrating schistosomiasis control into broader climate resilience planning.

By consolidating and analyzing research on these topics, this review seeks to provide a comprehensive overview of the connections between climate change, schistosomiasis transmission, and public health strategies. The findings will help to clarify how climate-induced environmental changes may alter disease distribution, inform predictive models, and support more effective, integrated approaches to managing schistosomiasis in the face of global climate change.

This scoping review adds significant value to the current body of literature by offering a comprehensive synthesis of how multiple climate variables, such as temperature, rainfall, and water body dynamics, jointly influence the ecology of snail vectors and the transmission of schistosomiasis. Unlike previous studies that often focus on single climatic factors or isolated regions, this review integrates diverse environmental drivers to present a more holistic understanding of disease risk under climate change. It is among the few reviews to identify emerging schistosomiasis hotspots in regions previously considered non-endemic and to highlight how climate change may shift or contract endemic zones, particularly emphasizing areas like East Africa, Southeast Asia, and parts of South America. Furthermore, the review distinguishes itself by systematically evaluating the role of climate adaptation policies, including water resource management, sanitation improvements, and early warning systems in mitigating transmission. It also fills a critical gap in linking climate science with public health strategies by advocating for the integration of climate adaptation into schistosomiasis control programs, especially in regions facing increased vulnerability due to climate variability.

## 2. Methodology

This scoping review aims to evaluate the impact of climate change on the transmission and distribution of schistosomiasis, with a particular focus on how environmental factors such as rainfall, temperature, and water bodies influence snail populations, and how these changes affect the disease dynamics. This review assesses shifts in endemic areas and the implications for disease control efforts in the context of climate change adaptation policies. The methodology followed the Arksey and O’Malley framework [27], supplemented by Levac et al.’s enhancements [28], and adhered to the PRISMA-ScR guidelines [29] to ensure a rigorous and transparent review process. This review protocol was registered with the Open Science Framework (OSF), hosted by the Center for Open Science (COS), Washington DC, under the registration link; https://doi.org/10.17605/OSF.IO/EYXCS on 3 December 2024.

### 2.1. Research Questions

The primary objective of the scoping review was to explore the various ways in which climate change influences the transmission and distribution of schistosomiasis, with a focus on the environmental factors that alter transmission dynamics and the broader implications for public health. Specifically, the review addresses the following research questions: (1) How do changing rainfall patterns, temperature fluctuations, and alterations in water bodies affect snail populations and the transmission dynamics of schistosomiasis?; (2) How have schistosomiasis-endemic areas shifted or are predicted to shift due to climate change, and which regions are emerging as future hotspots for transmission?; and (3) What is the impact of climate adaptation policies on schistosomiasis control efforts, and how can these policies be integrated into public health strategies? These questions were designed to provide insights into the complex relationship between climate change and schistosomiasis transmission, highlighting areas of concern for public health and guiding policy interventions.

### 2.2. Eligibility Criteria

To ensure the relevance and focus of the review, the eligibility criteria included studies published between 2000 and 2024 that investigate the impact of climate change on schistosomiasis transmission. Eligible studies that focused on environmental factors such as temperature, rainfall, and water bodies and their effects on the life cycle of schistosomiasis, particularly snail vectors (e.g., *Biomphalaria*, *Bulinus*). Both quantitative and qualitative studies were considered, provided they present data on changes in transmission dynamics, shifts in endemic areas, or the effectiveness of climate adaptation strategies in controlling the disease. Studies that reported empirical evidence on climate models, future transmission predictions, or the implementation and outcomes of climate adaptation policies were included. Only English-language studies or those with English translations were included to facilitate data analysis. Exclusion criteria were applied to studies that do not directly address climate change, focus on other diseases, or are opinion-based without empirical data. These criteria helped in filtering studies that directly address the intersection of climate change and schistosomiasis transmission dynamics.

### 2.3. Identification of Relevant Studies

A comprehensive search strategy was employed across multiple databases to capture relevant studies. The primary databases searched included PubMed, Scopus, Web of Science, and Google Scholar, which cover a wide range of ecological, environmental, and health sciences literature. Additionally, grey literature sources, such as reports from the World Health Organization (WHO), the United Nations, and other climate and health organizations, were reviewed to include data from governmental and non-governmental organizations that address climate change impacts on schistosomiasis. The search was done using a combination of controlled vocabulary (e.g., MeSH terms) and free-text keywords, including “climate change schistosomiasis”, “snail population and climate”, “temperature and schistosomiasis transmission”, “rainfall patterns schistosomiasis”, “climate adaptation policies and schistosomiasis”, and “future transmission hotspots”. Boolean operators (AND, OR) were applied to combine these terms effectively and expand the search scope. This approach ensured that studies related to the environmental factors influencing schistosomiasis transmission were comprehensively captured.

### 2.4. Study Selection Process

The study selection process was conducted in two stages. In the first stage, two independent reviewers screened the titles and abstracts of all retrieved studies against the pre-defined eligibility criteria. Studies that met the initial screening criteria proceeded to the second stage, where full-text articles were reviewed by the same two independent reviewers for inclusion. A third reviewer resolved any disagreements in the study selection. Reasons for exclusion at both stages were documented to maintain transparency in the process. The PRISMA flow diagram was used to track the screening and selection process, ensuring that the methodology was documented. The two-stage selection process allowed for a thorough examination of studies and ensured that only the most relevant studies were included in the review.

### 2.5. Data Extraction, Synthesis and Analysis of Results

Data extraction followed a standardized format to capture study and population characteristics, environmental factors (rainfall, temperature, water body changes), snail population dynamics, transmission trends, and the effectiveness of climate adaptation strategies. Priority was given to studies examining snail ecology and climate-driven shifts in schistosomiasis transmission. Extracted data were organized using spreadsheets or databases and analyzed through thematic and descriptive synthesis. This included mapping temporal and geographical changes in disease transmission, evaluating the ecological drivers affecting snail populations, and assessing the effectiveness of climate adaptation policies. The synthesis highlighted key findings, research gaps, and future directions, especially around predictive modeling and integrated control strategies.

## 3. Results

### 3.1. Study Characteristics

The study involved a comprehensive search across four electronic databases: Scopus, PubMed, Google Scholar, and Web of Science. The aim was to identify literature on climate change and the transmission dynamics of schistosomiasis, shifts in endemicity, the emergence of new and potential transmission hotspots, and the impact of climate-adapted policies on schistosomiasis control efforts. This search yielded a total of 24,614 articles. After removing duplicates and screening titles and abstracts, 98 full-text articles met the inclusion criteria. An additional search for grey literature, including theses, policy documents, and official releases, identified 23 more relevant sources. Altogether, 121 studies and reports were included in the analysis (Figure 1) [11,12,13,15,16,19,22,24,30,31,32,33,34,35,36,37,38,39,40,41,42,43,44,45,46,47,48,49,50,51,52,53,54,55,56,57,58,59,60,61,62,63,64,65,66,67,68,69,70,71,72,73,74,75,76,77,78,79,80,81,82,83,84,85,86,87,88,89,90,91,92,93,94,95,96,97,98,99,100,101,102,103,104,105,106,107,108,109,110,111,112,113,114,115,116,117,118,119,120,121,122,123,124,125,126,127,128,129,130,131,132,133,134,135,136,137,138,139,140,141,142,143].

### 3.2. Environmental Changes and Schistosomiasis Transmission

Shifts in rainfall, temperature, and water body conditions directly impact snail populations, influencing the spread of schistosomiasis. These changes create favorable habitats for snails, altering disease transmission dynamics and increasing the risk of infection in affected areas. The reports on the impact of climate change on schistosomiasis transmission that focus on how shifting environmental conditions could affect endemic areas and create new hotspots were systematically retrieved and analyzed. The findings suggest that climate change is likely to alter the geographical distribution of schistosomiasis, with some regions experiencing increased *Schistosoma* transmission risk [30,32,33,36]. Areas that were previously not suitable for transmission, such as parts of Southeast Asia [43,44,45,46], the Pacific Islands [47,72] and Mediterranean Region [60,61,62,63], and the Amazon Basin [33,47,48], are expected to become new hotspots due to changes in temperature and rainfall patterns. In contrast, regions that are already near or at the optimal temperature for transmission, like parts of West Africa, may see a reduction in risk as temperatures rise further [12,23,41,42] (Table 1). The environmental factors, including temperature fluctuations, rainfall, and water body changes, influence snail populations (*Biomphalaria*, *Bulinus*, and *Oncomelania*) and, subsequently, the transmission dynamics of schistosomiasis [30,31,32,33,34,35,36,37]. As temperatures and precipitation increase, snail populations may rise, leading to a higher risk of transmission [30,32,36]. Conversely, transmission may decrease in areas where temperatures exceed the thermal optimum for snail survival [37]. The implications of these findings suggest that public health strategies will need to adapt to these shifting dynamics, with a focus on surveillance and early intervention in emerging hotspots [30,31,32,33,34,35,36,37]. These underscore the need for climate health models to predict the future spread of schistosomiasis and inform targeted public health responses. The climate adaptation policies and schistosomiasis control are shown in Table 2.

### 3.3. How Environmental Factors Influence Schistosoma Infection in Freshwater Snails

Environmental factors such as extreme weather, floods, temperature, and rainfall significantly impact *Schistosoma* infection rates in freshwater snails. Favorable conditions like high humidity, warm temperatures, and expanded wetland areas increase snail infection rates, driving disease transmission. Conversely, factors like water scarcity and temperature shifts can reduce infections. These environmental factors significantly affect *Schistosoma* infection rates among intermediate freshwater snails. Extreme weather patterns emerged as the most influential factor, creating new transmission risks and leading to a high infection rate of 0.6 [35]. Floods contributed to larger wetland areas conducive to snail survival and increased humidity, with a comparable infection rate of 0.5 [36]. Similarly, temperature conditions favoring snail growth also correlated with an infection rate of 0.5, while low rainfall and high temperature resulted in a slightly reduced rate of 0.3 [37]. Conversely, factors like temperature shifts that disrupt snail activity cycles significantly reduced the infection rate to 0.1 [33]. Other factors, including water scarcity (0.3) [31] and consistent rainfall (0.35) [34], also exhibited a dampening effect on snail schistosomiasis infections (Figure 2). These findings highlight the intricate interplay between environmental changes and disease transmission risks, underscoring the importance of environmental monitoring in schistosomiasis control programs.

### 3.4. Countries with High Schistosoma Infection Rates in Snails

Environmental changes have contributed to the emergence of new schistosomiasis hotspots in various countries, significantly affecting the infection rates among intermediate freshwater snails. These variations indicate the role of localized environmental factors, such as flooding, temperature shifts, and humidity changes, in creating favorable conditions for snail survival and *Schistosoma* transmission (Figure 3).

### 3.5. Link Between Snail Infection Rates and Human Schistosomiasis

Higher *Schistosoma* infection rates in freshwater snails are closely linked to increased schistosomiasis cases in humans, as snails amplify the transmission cycle when environmental factors favor their survival. Mozambique leads with the highest infection rate at 0.6 [35], followed closely by Nigeria 0.5 [36] and Ghana 0.5 [32]. Egypt reported a moderately high rate of 0.4 [37], while Tanzania records 0.35 [34]. For example, Mozambique’s highest snail infection rate 0.6 corresponds to a 20–25% human exposure to schistosomiasis in rural areas [35]. Similarly, Ghana reported a 20% increase in human infections in affected regions [32], while Nigeria observed a 15–18% rise in communities near water bodies [36]. Interestingly, even low snail infection rates, such as 0.12, can result in 25.6% human infection in high-exposure areas, as seen in Uganda [30] (Figure 4).

### 3.6. Climate Change and Emerging Schistosomiasis Hotspots

Climate change is expected to shift schistosomiasis-endemic areas, creating new hotspots for transmission. Temperature (Figure 5a), rainfall (Figure 5b), and humidity changes may expand snail habitats, increasing the risk of infection in previously unaffected regions and intensifying transmission in existing areas. The results of this study show a significant shift in schistosomiasis transmission (Figure 5c) patterns due to climate change. Of the 73 studies published between 2000 and 2024 focused on the impacts of climate-induced changes in temperature, rainfall, and water bodies on schistosomiasis-endemic regions [11,12,13,15,16,19,24,33,38,39,40,41,42,43,44,45,46,47,48,49,50,51,52,53,54,55,56,57,58,59,60,61,62,63,64,65,66,67,68,69,70,71,72,73,74,75,76,77,78,79,80,81,82,83,84,85,86,87,88,89,90,91,92,93,94,95,96,97,98,99,100,101,102,103,104]. The majority of the studies 91.8% (58/73) indicated that climate change would likely increase transmission in regions where schistosomiasis is already endemic [11,12,15,16,24,38,39,40,43,44,45,46,47,48,49,50,51,52,53,58,59,60,61,62,63,64,65,66,67,70,71,72,73,74,76,77,78,79,80,81,84,85,86,87,88,89,90,91,92,95,96,97]. In Sub-Saharan Africa [11,16,38,39], Southeast Asia [43,44,45,46], and parts of South America [33,47,48], higher temperatures and altered rainfall patterns were found to expand suitable habitats for snail vectors, potentially lengthening transmission seasons and increasing infection rates in both snails and humans. For example, regions in East Africa, such as Ethiopia and Sudan [79,80,81], were predicted to experience longer wet seasons, thus extending transmission periods. Similarly, in Southeast Asia, countries like Cambodia [84,85,88,89,90] and Indonesia [76,77,78] were expected to see increased transmission risk due to intensified monsoon seasons and rising temperatures, which would enhance the survival and reproduction of snails that harbour schistosomiasis parasites. Interestingly, WHO data maps illustrating annual schistosomiasis prevalence trends across different regions highlight significant shifts in disease burden over time to the previously nonendemic regions (Figure 6). In contrast, in some regions, 17.8% (13/73) were expected to experience a decrease in transmission risk due to climate change [12,33,41,42,75,82,83,93,94,101,102,103,104]. Higher temperatures and reduced rainfall in parts of West Africa [12,33,41,42], particularly in the Sahel and coastal zones, could limit snail habitats, leading to a reduction in transmission. The Mediterranean and Middle Eastern regions, including Türkiye and Greece [93,94], were also predicted to see reduced transmission as hotter and drier conditions become less favorable for snail populations. Additionally, several studies identified emerging hotspots for schistosomiasis transmission, particularly in regions that are not currently endemic but are predicted to become more favorable for transmission due to shifts in climate. Areas such as Kenya [38,65,66], Tanzania [67], Zambia [58], and parts of Southeast Asia, like Myanmar [88,89] and Laos [75], are likely to experience increased transmission as rainfall and temperature changes create more suitable environments for snails. In South America, the Amazon basin and northern regions of Brazil and Colombia were flagged as potential new hotspots [33,47,48], where altered rainfall patterns and increased temperatures are expected to raise transmission risk.

### 3.7. Climate Adaptation Policies and Schistosomiasis Control

The studies collectively emphasized the importance of integrating climate change models into schistosomiasis control programs, which would enable more effective predictions of transmission risks and inform tailored interventions (Table 2).

The findings suggest an urgent need to adapt public health strategies to address the evolving transmission dynamics of schistosomiasis in the context of climate change. With climate change influencing both the expansion of endemic regions and the emergence of new hotspots, public health systems must enhance surveillance, adjust control measures, and focus efforts on emerging regions where the disease may spread. Furthermore, regional collaboration and climate-sensitive interventions were essential to mitigate the public health impact of these shifts in schistosomiasis transmission.

Climate adaptation policies play a vital role in schistosomiasis control by addressing environmental factors like flooding, temperature changes, and water scarcity that influence snail habitats and disease transmission. Integrating these policies into public health strategies can enhance disease control by promoting flood management, improved water infrastructure, and targeted snail control programs. Combining climate resilience with public health initiatives ensures sustainable interventions to reduce schistosomiasis transmission in vulnerable communities. The results from the 42 studies included in this review highlight the significant role of climate adaptation policies in enhancing schistosomiasis control efforts [105,106,107,108,109,110,111,112,113,114,115,116,117,118,119,120,121,122,123,124,125,126,127,128,129,130,131,132,133,134,135,136,137,138,139,140,141,142,143]. One of the most effective strategies identified was climate-resilient water management [106,113,128,136,137], particularly flood control [119] and irrigation management [136], which were shown to reduce the creation of suitable habitats for snail populations, the primary vectors in schistosomiasis transmission. Studies in Senegal [138], Tanzania [143], and Zambia [137] revealed that managing flood risks and improving irrigation systems could significantly decrease transmission rates. In addition, improving sanitation and water quality emerged as another critical intervention, particularly in areas with limited access to clean water. Countries such as Zambia [110,137], Ghana [129], and Uganda [127], which invested in sanitation infrastructure and wastewater management, experienced a marked reduction in human infections and snail populations.

Implementing an early warning system for climate extremes such as floods and droughts was crucial in preventing schistosomiasis outbreaks [108,135]. These systems enabled real-time responses to climate-induced risks, allowing for the distribution of preventive measures and public health interventions. Integrating climate adaptation policies into public health systems, particularly in regions vulnerable to climate change, was essential for maintaining sustainable schistosomiasis control programs [115,118,125,135,142]. Strengthening health system resilience, through capacity building and climate-responsive health infrastructure, was also a key recommendation from several studies [125,141,142].

Moreover, community-driven initiatives and climate-smart agricultural practices were effective in reducing transmission rates [36,109,119,122]. By combining water resource management with community engagement in regions like Ghana [129,133] and Zambia [110,137], local populations were empowered to control snail habitats while enhancing their resilience to climate change. The importance of multi-sectoral collaboration was another common theme across the studies, with successful outcomes resulting from coordinated efforts between climate, water management, and public health authorities.

## 4. Discussion

The findings from this review highlight the intricate relationship between climate change and the transmission dynamics of schistosomiasis [16,19,33]. As global temperatures rise and rainfall patterns shift, the geographical distribution of schistosomiasis is expected to change, with both endemic regions and new areas potentially seeing altered transmission dynamics [17]. Environmental factors such as temperature, precipitation, and water availability directly influence freshwater snail hosts’ survival, reproduction, and distribution, which are central to the transmission cycle [11,12,33,144]. These findings align with previous studies that have shown how environmental changes, driven by climate change, can increase the availability of habitats suitable for snails, thus creating more favorable conditions for schistosomiasis transmission [145,146].

The evidence from this study suggests that schistosomiasis transmission is likely to increase in regions where the disease is already endemic, especially in parts of Sub-Saharan Africa [38,39], Southeast Asia [43,52,53,88,89,90], and South America [33,47,48]. Changes in temperature and altered rainfall patterns in these regions are expanding habitats for snail populations, extending transmission seasons and increasing human exposure to the parasites [17,18,21,22]. For instance, in East Africa, longer wet seasons could extend transmission periods [11,19,38,50], while Southeast Asia, with intensified monsoon seasons [43,44,45,46,51,52,53,64], is predicted to see a rise in snail populations, thereby increasing transmission risk. This finding supports an earlier study that highlighted the role of warmer temperatures and increased rainfall in accelerating snail breeding cycles, potentially leading to higher infection rates [147].

In contrast, this review also points to some regions where climate change may reduce transmission risk [12,33,41,42,75,82,83,93,94,101,102,103,104]. Areas that are currently on the edge of optimal temperature ranges for snail survival, such as parts of West Africa [12,33,41,42] and the Mediterranean [93,94], may experience a decline in transmission due to rising temperatures and reduced rainfall. These environmental shifts could limit snail habitats and shorten transmission seasons [11,17,33]. However, it is important to note that this decrease in transmission risk might be offset by human-driven factors such as irrigation practices or dam construction, which can inadvertently create new habitats for snails, as seen in the unintentional consequences of climate adaptation efforts.

The emerging hotspots identified in this review underscore the unpredictable nature of climate change’s impact on schistosomiasis transmission [75,76,77,78,79,80,81,82,83,84,85,86,87,88,89,90,91,92,93,94,95,96,97,98,99,100,101,102,103,104]. While areas such as Kenya [38,65,66], Tanzania [67], Zambia [58], Myanmar [88,89], Laos [75], and parts of the Amazon basin [47,48] may face increased transmission risk due to changes in rainfall and temperature, these regions are not currently endemic to schistosomiasis. As climate-induced environmental changes create more suitable conditions for snails, these regions may see a shift in disease burden, placing additional pressure on public health systems that may not be prepared to address this new risk. This highlights the importance of proactive monitoring and early intervention strategies in regions at risk of becoming new hotspots.

The role of climate adaptation policies is another critical aspect of this review. As governments and organizations implement climate adaptation and resilience strategies, there is an increasing need to consider their potential impact on schistosomiasis transmission. For example, water resource management practices such as the construction of dams, while essential for addressing water scarcity, can inadvertently create ideal breeding grounds for snails [148,149]. Similarly, agricultural practices like expanded irrigation may increase human exposure to snails if not carefully managed [25,150,151]. This review emphasizes the need for an integrated approach to climate adaptation and public health, where the potential impacts on disease transmission are evaluated and mitigated.

In line with these concerns, this study points to several key policy recommendations. One such recommendation is the need for comprehensive climate-health models that incorporate projections of climate-induced shifts in schistosomiasis transmission [152,153]. These models can help predict future hotspots, guide surveillance efforts, and inform targeted interventions. Additionally, the study stresses the importance of integrating climate adaptation policies into schistosomiasis control programs, ensuring that both climate resilience and disease prevention are prioritized simultaneously [115,136,141]. Such integration requires close collaboration between climate, public health, and water management authorities to develop policies that are climate-sensitive and health-conscious [154,155,156].

Furthermore, community-driven initiatives and localized climate-smart agricultural practices can play a key role in reducing transmission [106,109,110,114,123,127,129,133,137,141,142]. In countries like Ghana [109,129,133,142], Zambia [110,137], and Uganda [106,114,123,127,141], successful implementation of these strategies has been linked with a reduction in the schistosomiasis burden. These community-driven initiatives are not specifically targeted at schistosomiasis-endemic communities, leading to the continued neglect of the disease. This oversight persists despite growing evidence that such interventions, particularly those focused on water, sanitation, and environmental management, can significantly reduce schistosomiasis transmission when implemented effectively. These community-based approaches, combined with effective water management and infrastructure improvements, empower local populations to manage snail habitats while also adapting to climate change [36,119,122,127,131,139,140]. The findings underscore the importance of multi-sectoral collaboration to ensure that interventions are not only effective but also sustainable in the long term.

The COVID-19 lockdowns in 2020 and 2021 offer a compelling example of how reduced human activity can alter environmental dynamics and, potentially, disease transmission patterns. During these periods, countries worldwide reported significant improvements in water and air quality, reductions in industrial emissions, and changes in land use pressures. Indeed, CO_2_ emission was drastically reduced during the COVID-19 pandemic due to the diminished activity, but if this influences the climate change/rise of temperature globally cannot be evaluated. While the primary focus of global attention was understandably on pandemic containment, these unintended environmental effects highlighted the sensitivity of ecological systems to anthropogenic activity. Though limited empirical data exist specifically linking the lockdowns to shifts in schistosomiasis transmission, the broader implications are clear: reduced pollution, less water contamination, and slowed infrastructure development may have temporarily influenced/modified the creation of snail habitats, which possibly favors snail survival and propagation. This period underscores the critical role human activity plays in shaping disease ecologies and reinforces the importance of sustainable, climate-smart policies. Integrating these lessons into climate adaptation planning could provide dual benefits for environmental restoration and infectious disease control, including neglected tropical diseases like schistosomiasis.

## 5. Limitations of the Study

This study has several limitations, including geographic and data bias, focusing primarily on regions with existing schistosomiasis data while neglecting areas with emerging risks or insufficient information. Interestingly, a review highlights schistosomiasis, once confined to tropical regions, as an emerging public health concern in Europe due to climate change expanding the habitat of snail vectors, with Southern Europe at risk of increased local transmission [157]. It oversimplifies the complex interactions between climate factors and human-driven influences, such as host immunity [158,159], irrigation and urbanization, and fails to fully capture the ecological dynamics of the disease. Uncertainties in climate model projections and a lack of integration of socioeconomic and population movement factors further limit predictive accuracy.

The generalizability of findings is constrained, as climate change impacts vary across regions due to differing environmental and health system contexts. The focus on snail-related factors underrepresents other transmission dynamics, such as human behavior and parasite survival. Policy evaluation is limited, with insufficient evidence on the effectiveness and scalability of climate adaptation strategies in controlling schistosomiasis. Predictions about emerging hotspots lack empirical validation, and the study does not address the impact of co-infections or interactions with other diseases. Addressing these gaps requires longitudinal research, improved models, and comprehensive policy assessments.

## 6. Future Research

Future research should prioritize longitudinal studies on the infectivity of snail intermediate hosts with *Schistosoma* and their capacity for sustained transmission to humans, to better understand the long-term impacts of climate change on schistosomiasis. Given the expanding geographic range of the disease, close monitoring of the presence or emergence of competent snail vectors in non-endemic areas is essential. This should involve active environmental surveillance for *Schistosoma* cercariae in water bodies, which serve as a direct indicator of transmission risk. These ecological monitoring efforts must be complemented by public health surveillance in human populations, including parasitological diagnostics and, where appropriate, serological tools such as antibody screening to detect past or ongoing exposure. Integrated surveillance strategies are crucial for early detection of emerging transmission foci and for guiding timely, targeted interventions in at-risk regions. Additionally, expanding data collection to underrepresented areas is critical for building a more globally representative evidence base. Improved climate-health models that incorporate environmental, socioeconomic, and human-driven factors are urgently needed to enhance the accuracy of transmission predictions. Finally, empirical validation of predicted hotspots and studies on schistosomiasis co-infections are necessary to refine risk assessments and understand broader health impacts.

Policy-oriented research should evaluate the effectiveness and scalability of climate adaptation strategies, such as water resource management and agricultural practices, in reducing transmission risks. Strengthening community-driven initiatives and fostering collaboration between climate, public health, and policy sectors were critical for developing sustainable, integrated solutions to mitigate the effects of climate change on schistosomiasis.

## 7. Conclusions

This review emphasizes the importance of integrating climate change projections into schistosomiasis control strategies. The potential for climate-induced shifts in schistosomiasis transmission highlights the need for public health systems to adapt to these changes through enhanced surveillance, predictive modelling, and targeted interventions in emerging hotspots. By combining climate adaptation strategies with schistosomiasis control efforts, public health programs can reduce the burden of this neglected tropical disease and better prepare for the future challenges posed by climate change.

## Figures and Tables

**Figure 1 ijerph-22-00812-f001:**
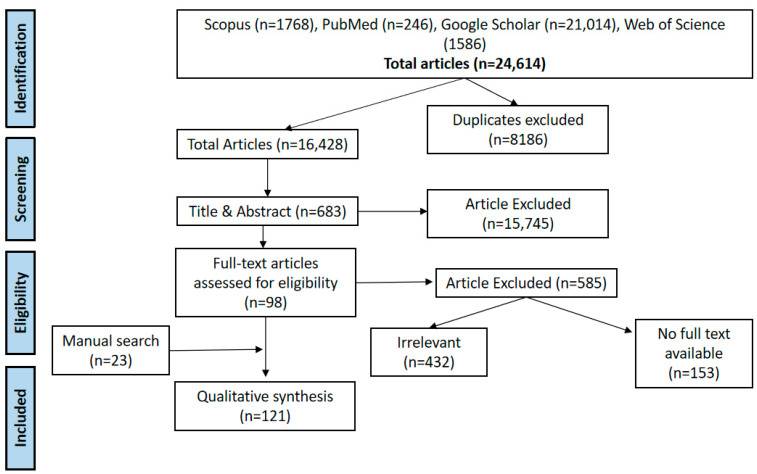
PRISMA flow chart for search and selection of included studies.

**Figure 2 ijerph-22-00812-f002:**
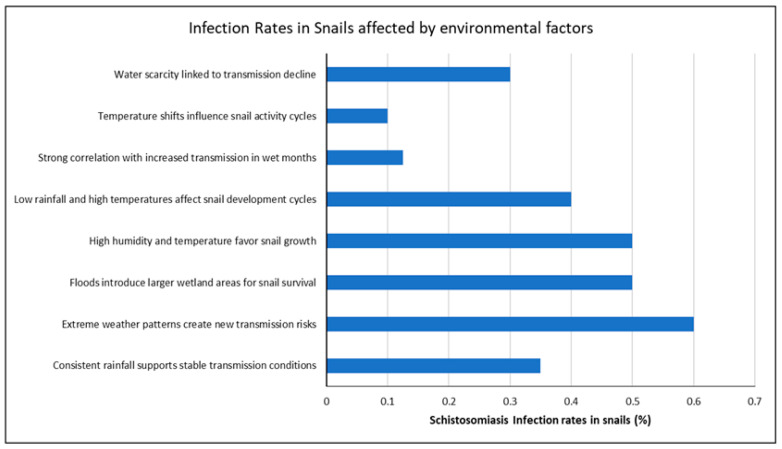
*Schistosoma* infection rates in snails influenced by environmental factors.

**Figure 3 ijerph-22-00812-f003:**
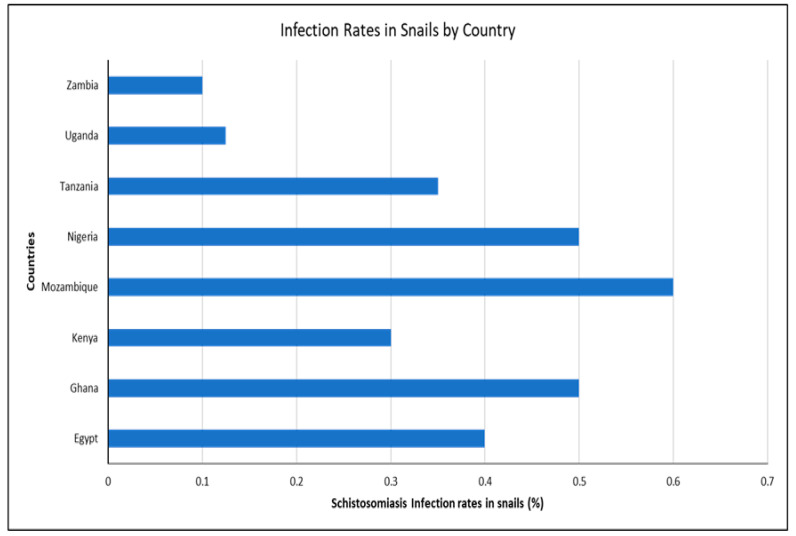
*Schistosoma* infection rates in snails across countries.

**Figure 4 ijerph-22-00812-f004:**
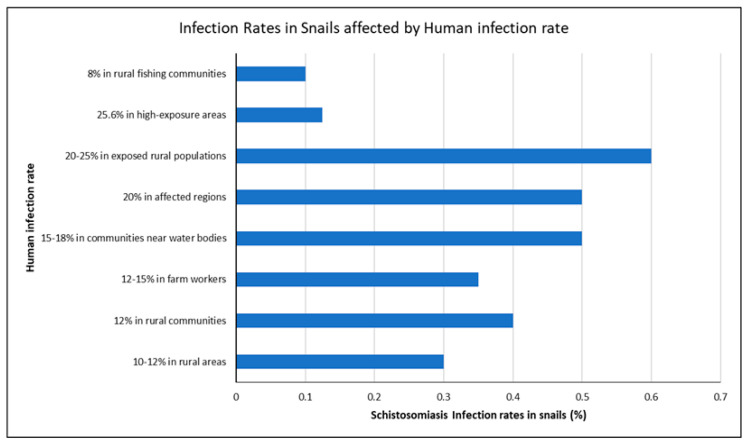
*Schistosoma* infection rates in snails and schistosomiasis prevalence in humans.

**Figure 5 ijerph-22-00812-f005:**
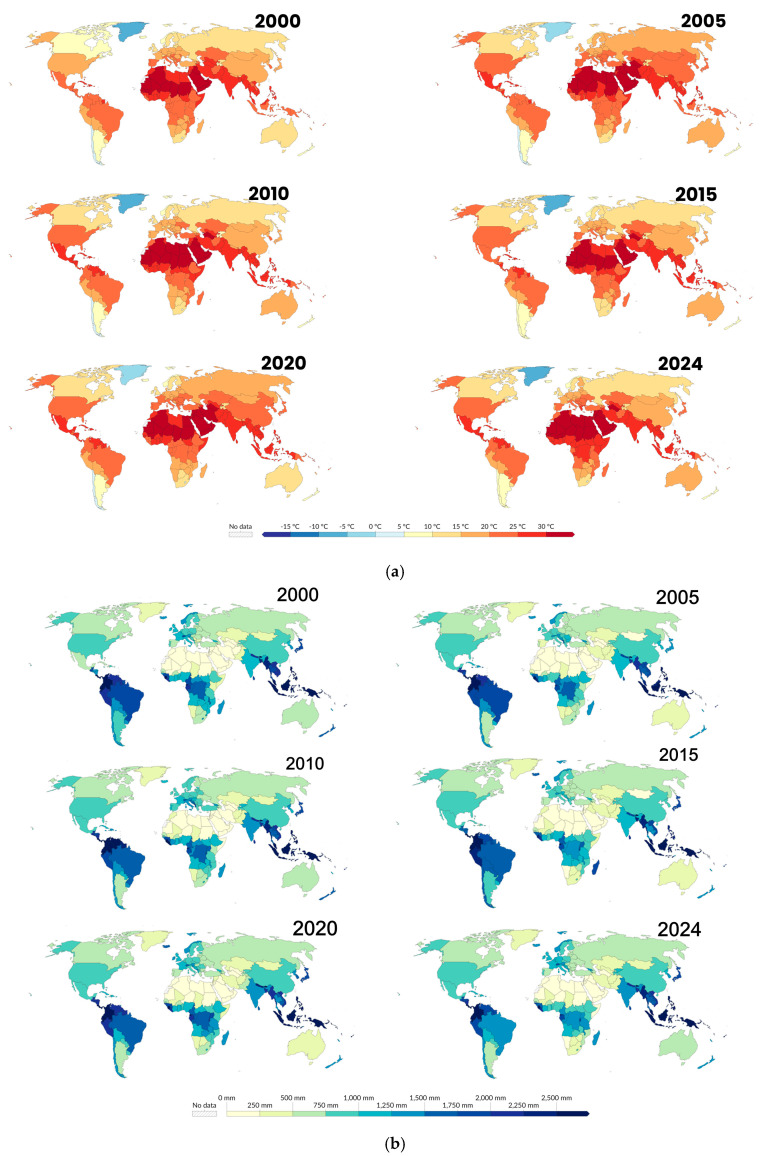
Global pattern of climate change and schistosomiasis prevalence. (**a**) Average monthly surface temperature changes from 2000 to 2024. https://ourworldindata.org/grapher/average-monthly-surface-temperature?time=2019-11-15&country=~GHA (accessed on 17 January 2025). (**b**) Changes in annual precipitation. https://ourworldindata.org/grapher/average-precipitation-per-year?time=2024 (accessed on 17 January 2025). (**c**) Patterns of annual schistosomiasis prevalence and treatment coverage. https://ourworldindata.org/grapher/schistosomiasis-treatment-coverage?time=2022 (accessed on 19 January 2025).

**Figure 6 ijerph-22-00812-f006:**
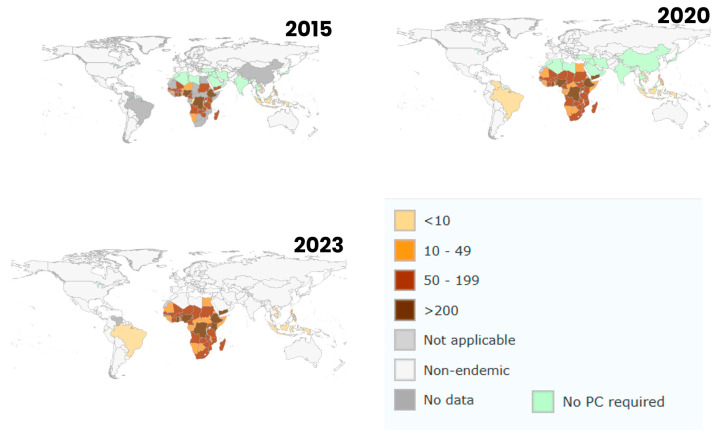
Changing patterns of global schistosomiasis prevalence from 2015 to 2023. The data maps illustrate annual prevalence trends across different regions, highlighting shifts in disease burden over time. Data were obtained from the World Health Organization (WHO), which aggregates yearly reports from national, regional, and global surveillance systems. https://apps.who.int/neglected_diseases/ntddata/sch/sch.html (accessed on 19 January 2025).

**Table 1 ijerph-22-00812-t001:** Climate change and emerging schistosomiasis hotspots.

Region	Current Transmission Suitability	Predicted Impact of Climate Change	Future Hotspot	Key Factors	Potential Effects	Infection Rates (Snails)	Human Infection Rates	Key Findings	References
Sub-Saharan Africa	High	Increased transmission risk	Eastern Africa	Temperature rise, seasonal rainfall	Higher transmission during wet seasons	High	Increased in wet seasons	Climate change leads to more favorable conditions for snail reproduction in wetter areas	[11,16,38,39]
Southern Africa	Moderate	Increased transmission risk	Southern Africa	Increased temperatures, higher rainfall	Longer transmission seasons	Moderate	Moderate	Increased rainfall and temperature increase snail density, extending transmission seasons	[15,24,40]
West Africa	Moderate	Decreased transmission risk	West Africa	Temperature rise, water body changes	Decreased transmission risk due to temperature rise	Low	Low	Higher temperatures limit snail survival, reducing transmission risk	[12,33,41,42]
Southeast Asia	Low	Increased transmission risk	Southeast Asia	Monsoon, temperature increase	Increased transmission risk due to wet season	Low	Low	Monsoon shifts could introduce new areas for snail host survival	[43,44,45,46]
South America	Low	Increased transmission risk	Northern South America	Rainfall, temperature rise	Increased risk due to seasonal flooding	Low	Low	Changes in rainfall patterns increase areas of suitable habitat for snails	[33,47,48]
Central Africa	High	Increased transmission risk	Central Africa	Higher temperatures, extended rainy season	Snail population densities increase in warmer months	High	Increased in wet season	Climate-driven increases in water bodies lead to higher snail populations and infection rates	[12,16,22,49]
East Africa	Moderate	Increased transmission risk	Eastern Africa	Changing rainfall patterns, rising temperatures	Prolonged wet season increases transmission risk	Moderate	Increased	Increased rainfall leads to an extended breeding period for snails	[11,19,38,50]
South East Asia	Low	Increased transmission risk	Southeast Asia	Higher monsoon intensity, rising temperatures	Increased snail populations in monsoon season	Low	Low	Rising temperatures and monsoon lengthening transmission windows	[51,52,53]
Central Asia	Low	No significant change	Central Asia	Mild temperature fluctuations	Stable snail populations in mild conditions	Low	Low	Little to no impact from climate change in stable, low-transmission regions	[43,54,55,56]
Western Africa	Moderate	Increased transmission risk	Western Africa	Rainfall variation, temperature increases	Snail population peak during rainy season	Moderate	Moderate	Variable rainfall patterns increase risks for schistosomiasis transmission	[13,57,58,59]
Mediterranean Region	Low	Increased transmission risk	Mediterranean countries	Temperature rise, water body alterations	Longer transmission seasons due to higher temperatures	Low	Low	Shifting water bodies due to temperature changes could introduce new transmission foci	[60,61,62,63]
Southern Asia	Moderate	Increased transmission risk	South Asia	Increased monsoons, rising temperatures	Increased snail populations during monsoon periods	Moderate	Moderate	Extended monsoon season likely increases snail populations, raising transmission risk	[43,64]
East Africa	High	Increased transmission risk	Kenya, Tanzania	Higher temperatures, reduced water bodies	Decreased transmission in drier periods	High	Increased in wet seasons	Drying water bodies may reduce snail habitats in some areas, but wet season populations may spike	[38,65,66,67]
Caribbean Islands	Low	No significant change	Caribbean islands	Stable temperature, rainfall fluctuations	Stable transmission rates	Low	Low	Stable environmental conditions limit changes to transmission risk	[68,69]
Central America	Moderate	Increased transmission risk	Central America	Rising temperatures, changing rainfall patterns	Longer wet season extends transmission risks	Moderate	Increased	Lengthened wet season extends periods of snail-host availability	[70,71]
Pacific Islands	Low	Increased transmission risk	Pacific Islands	Rising sea levels, increased rainfall	Increased transmission in flooded areas	Low	Low	Flooded areas may support new snail populations, raising infection risks	[44,72]
West Africa	High	Increased transmission risk	West Africa	Longer rainy season, rising temperatures	Increased transmission during wet periods	High	Increased in wet seasons	Longer rainy season increases snail-host survival, leading to increased infection rates	[73,74]
South East Asia	Moderate	Decreased transmission risk	Vietnam, Laos	Temperature fluctuations, rainfall variability	Shorter wet season reduces transmission risk	Moderate	Low	Shorter wet season may limit available breeding conditions for snails	[75]
Tropical Asia	Moderate	Increased transmission risk	Indonesia, Philippines	Rising temperature, longer wet season	Extended breeding period for snails	Moderate	High	Extended rainy season and warmer temperatures increase risks for both snail population and human rates	[76,77,78]
Middle East	Low	No significant change	Middle East	Stable temperatures, periodic rainfall	Stable transmission levels	Low	Low	Dry conditions and stable water levels result in minimal change to transmission	[62]
East Africa	High	Increased transmission risk	Ethiopia, Sudan	Temperature rise, flooding	Snail population densities increase during floods	High	Increased in wet seasons	Flooding due to higher rainfall extends snail-host habitats, raising transmission risk	[79,80,81]
Central Africa	Moderate	Decreased transmission risk	Cameroon, Central African Republic	Temperature increase, dry conditions	Lower transmission due to lack of water bodies	Moderate	Low	Higher temperatures reduce available snail habitats, decreasing infection rates	[82,83]
South East Asia	High	Increased transmission risk	Thailand, Cambodia	Rising temperatures, extended wet season	Increase in snail populations and infection rates	High	Increased in wet season	Temperature rise extends breeding season for snails, increasing human transmission risk	[84,85]
East Africa	Moderate	Increased transmission risk	Uganda, Rwanda	Rainfall changes, temperature fluctuations	Transmission peak in rainy season	Moderate	High	Climate-induced rainfall shifts may extend wet season transmission period for schistosomiasis	[86,87]
Southeast Asia	Low	Increased transmission risk	Myanmar, Cambodia	Longer rainy season, rising temperature	Increased snail populations in new wetland areas	Low	Low	Seasonal flooding creates new areas for snail-host survival, raising transmission risk	[88,89,90]
West Africa	Moderate	Increased transmission risk	Nigeria, Ghana	Temperature rise, seasonal rainfall changes	Increased risk due to favorable breeding conditions	Moderate	Increased	Warmer wet season leads to increased snail density and infection risk	[41,91,92]
Eastern Mediterranean	Low	Decreased transmission risk	Türkiye, Greece	Rising temperatures, changing rainfall patterns	Reduced snail populations due to less favorable conditions	Low	Low	Rising temperatures reduce optimal snail habitats in the region	[93,94]
Central America	High	Increased transmission risk	Honduras, Panama	Increased rainfall, rising temperature	Higher transmission rates during wet season	High	Increased in wet season	Longer rainy periods contribute to increased snail population densities	[95,96,97]
East Asia	Low	No significant change	Japan, South Korea	Stable temperatures, minimal rainfall changes	Stable transmission levels	Low	Low	Stable environmental conditions lead to minimal impact on transmission risk	[98,99,100]
South Asia	Moderate	Decreased transmission risk	India, Bangladesh	Increased temperatures, seasonal rainfall	Shorter rainy seasons reduce transmission risk	Moderate	Low	Shorter rainy seasons may limit periods of transmission for schistosomiasis	[101,102,103,104]

**Table 2 ijerph-22-00812-t002:** Climate adaptation policies and schistosomiasis control.

Year	Location	Climate Adaptation Policy	Impact on Schistosomiasis Control	Policy Integration with Public Health	Key Findings	Implications for Schistosomiasis Transmission	References
2012–2014	Mozambique	Sustainable Irrigation, Water Supply, and Sanitation for Climate Challenges	Climate-smart irrigation lowers snail populations; improved rural water systems reduce contamination	Incorporated into irrigation policy frameworks and community health initiatives.	Improved irrigation management decreases snail habitats and schistosomiasis prevalence.	Enhanced sanitation reduces disease risk and prevents agricultural zone outbreaks.	[122,136]
2015–2017	Burkina Faso, Kenya; Uganda; Mozambique; Rwanda; Cameroon; Ghana; Mozambique	Integrated approaches including river basin planning, climate-resilient infrastructure, flood risk management, sustainable agriculture, urban resilience, and coordinated water and sanitation strategies.	Controlling water accumulation through river management, improved storage, drainage, and floodplain management reduces snail breeding and strengthens climate and health resilience.	Climate and health resilience measures are integrated across national river management, flood risk, water distribution, agricultural, urban planning, and health emergency systems.	Significant reductions in snail populations (up to 25%) and schistosomiasis transmission, especially in flood-prone areas, through improved sanitation and management.	Adapted health infrastructure, sustainable farming, and water management strategies reduce seasonal transmission spikes, prevent outbreaks, and control disease spread in urban and rural areas.	[111,112,123,124,126,128,129,141]
2018–2020	Uganda; South Sudan; Ethiopia; Senegal; Sierra Leone; Liberia; Tanzania; Democratic Republic of Congo	Comprehensive strategies for climate resilience, including water purification, flood management, disaster risk reduction, and climate-adapted infrastructure	Community-based water purification, flood protection, and climate-adapted systems reduce contamination risks and prevent snail population growth, ensuring better public health outcomes.	Integrated strategies linking local health, disaster management, food security, and community-based risk programs to enhance resilience and health outcomes	Snail population reduced by up to 40%, leading to decreased transmission and infection rates, especially in rural, flood-prone, and coastal areas, with improved healthcare access during climate extremes.	Water purification, flood management, and resilient infrastructure reduce transmission risks and prevent schistosomiasis outbreaks, particularly in vulnerable regions.	[117,120,125,127,131,134,138,139]
2021–2023	Kenya; Uganda; Liberia; Ghana; Zambia; Ethiopia; Sierra Leone; Nigeria; Tanzania; Democratic Republic of Congo; Senegal; Malawi	Integrated strategies for climate resilience, including water resource management, flood prevention, climate-smart agriculture, sanitation, and early warning systems.	Climate-resilient infrastructure, early flood warnings, improved water quality, and sanitation reduce snail habitats, prevent contamination, and lower schistosomiasis transmission.	Integrated strategies linking water, sanitation, agriculture, public health, and disaster response to enhance community health resilience and reduce disease risks.	Targeted interventions reduce schistosomiasis cases and transmission by up to 40%, with significant reductions in vulnerable regions, improved access to treatment, and lower disease burden.	Policies and strategies, including flood management, early warnings, water quality improvement, and sanitation, significantly reduce schistosomiasis transmission, especially in rural, urban, and high-risk communities.	[36,105,106,109,110,113,114,115,119,121,130,132,133,135,140,142,143]
2024	Tanzania; Malawi; Nigeria; Kenya; Zambia	Integrated strategies for climate resilience, including coastal adaptation, early warning systems, water management, and sanitation improvement.	Coastal defenses, water stagnation reduction, and improved sanitation reduce schistosomiasis transmission, with increased awareness and urban resilience policies enhancing disease control.	Integrated strategies linking national health, water management, disaster preparedness, and sanitation programs to enhance disease control and resilience.	Targeted interventions reduced schistosomiasis transmission by 30%, lowered snail populations, and decreased waterborne disease risk in agricultural, coastal, and urban areas.	Coastal and urban resilience policies, along with prevention measures and improved sanitation, play key roles in reducing schistosomiasis transmission.	[107,108,116,118,137]

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
