# Peer review of "Impact of Climate Change on Schistosomiasis Transmission and Distribution—Scoping Review"

_ijerph, 2025, doi:10.3390/ijerph22050812_

Round 1
Reviewer 1 Report
Comments and Suggestions for Authors
The manuscript explores an emerging field of research: the impact of climate change on neglected diseases, particularly schistosomiasis. The authors have conducted a literature review covering an extensive period from 2000 to 2024, which highlights their scientific efforts in this area.
Overall, the review is well-written, but I have some suggestions for improvement. Literature reviews and scoping reviews can often be tedious and may not adequately credit the authors due to their complexity. As a result, readers may find it challenging to engage with the paper and may lose interest. I encourage the authors to streamline the work for better readability.
Additionally, the number of tables in such a large review can be overwhelming and difficult to interpret. I suggest that the authors consider eliminating the first table and incorporating its information into the results section. The second table is well-structured and easy to read, while the third table could be improved by grouping the data into five-year intervals or by decades. This would enhance readability and better balance the length of the results with the discussion.
For instance, some figures could be replaced with reference links to reduce clutter. Furthermore, the discussion could be expanded to more closely reference the figures described, enriching the overall analysis.
In conclusion, I commend the authors for their significant scientific effort, which contributes valuable information to the scientific community.
Author Response
Reviewer 1
Comment 1; The manuscript explores an emerging field of research: the impact of climate change on neglected diseases, particularly schistosomiasis. The authors have conducted a literature review covering an extensive period from 2000 to 2024, which highlights their scientific efforts in this area.
Response: Thank you. We appreciate your comment.
Comment 2: Overall, the review is well-written, but I have some suggestions for improvement. Literature reviews and scoping reviews can often be tedious and may not adequately credit the authors due to their complexity. As a result, readers may find it challenging to engage with the paper and may lose interest. I encourage the authors to streamline the work for better readability.
Response: We have modified the manuscript based on the reviewer's comments.
Comment 3: Additionally, the number of tables in such a large review can be overwhelming and difficult to interpret. I suggest that the authors consider eliminating the first table and incorporating its information into the results section. The second table is well-structured and easy to read, while the third table could be improved by grouping the data into five-year intervals or by decades. This would enhance readability and better balance the length of the results with the discussion.
Response: We have deleted Table 1 and modified and changed Table 3 to Table 2 (line 271).
Comment 4: For instance, some figures could be replaced with reference links to reduce clutter. Furthermore, the discussion could be expanded to more closely reference the figures described, enriching the overall analysis.
Response: The figures were modified based on the data available and we also included the links to the data sources and the figures in the figure legends (see lines 360-364 and 370).
Comment 5: In conclusion, I commend the authors for their significant scientific effort, which contributes valuable information to the scientific community.
Response: Thank you.
Reviewer 2 Report
Comments and Suggestions for Authors
The authors attempted to review a comprehensive overview of the connections between climate change, schistosomiasis transmission, and public health strategies with the aim of clarifying how climate-induced environmental changes may alter disease distribution, inform predictive models, and support more effective, integrated approaches to managing schistosomiasis in the face of global climate change.
The work is comprehensively done with a sound methodology. The review helps clarify the connections between climate resilience initiatives and public health calling for a holistic integrated one health approach to climate change initiatives. This is very commendable.
However, the authors need to clarify a few things to make their work even better.
- Define prevalence and infection rates. Are these the same? Otherwise why use them interchangeably?
- There are populations that are more at risk than others for example those living near water bodies. How have they particularly been addressed?
- The tables are not very well presented and they could be improved.
Author Response
Reviewer 2
Comment 1: The authors attempted to review a comprehensive overview of the connections between climate change, schistosomiasis transmission, and public health strategies with the aim of clarifying how climate-induced environmental changes may alter disease distribution, inform predictive models, and support more effective, integrated approaches to managing schistosomiasis in the face of global climate change.The work is comprehensively done with a sound methodology. The review helps clarify the connections between climate resilience initiatives and public health calling for a holistic integrated one health approach to climate change initiatives. This is very commendable. However, the authors need to clarify a few things to make their work even better.
Response: Thank you for the comment.
Comment 2; Define prevalence and infection rates. Are these the same? Otherwise why use them interchangeably?
Response: Thank you for the question. Interestingly, there are slight differences between prevalence and infection rate. By definition, “Prevalence refers to the proportion of a population that has a particular disease or infection at a specific point in time or over a period of time” and “Infection rate usually refers to the rate at which new infections are occurring in a population over a specific period” (reference: Lucy A. McNamara, Stacey W. Martin, 1 - Principles of Epidemiology and Public Health,
Editor(s): Sarah S. Long, Charles G. Prober, Marc Fischer, Principles and Practice of Pediatric Infectious Diseases (Fifth Edition), Elsevier, 2018, Pages 1-9.e1, ISBN 9780323401814,https://doi.org/10.1016/B978-0-323-40181-4.00001-3. (https://www.sciencedirect.com/science/article/pii/B9780323401814000013)).We believe we used these words in their correct context.
Comment 3: There are populations that are more at risk than others for example those living near water bodies. How have they particularly been addressed?
Response: Yes, this is true, these community-driven initiatives are not specifically targeted at schistosomiasis-endemic communities, leading to the continued neglect of the disease. This oversight persists despite growing evidence that such interventions, particularly those focused on water, sanitation, and environmental management, can significantly reduce schistosomiasis transmission when implemented effectively (we have added this sentence to the discussion, see line 483-488).
Comment 4: The tables are not very well presented and they could be improved.
Response: We have modified the tables to increase readability. Table 1 has been deleted from the manuscript and Table 3 has been modified (Line 271).
Reviewer 3 Report
Comments and Suggestions for Authors
The manuscript entitled "Impact of Climate Change on Schistosomiasis Transmission and Distribution - Scoping Review”,examines the impact of climate change on schistosomiasis transmission and its implications for disease control. Despite the good structure of the manuscript, there are some concerns which should be addressed.
- I think no need to write a structured abstract as this is scoping review not research article.
- Pleas correct the word “Result” to be “Results”
- All tables; please design as Three-line tables and make them fit to page borders.
- Tables 1 and 3; can you reorder your content in an ascending years manner?
- Regarding figures; I think you need to provide high resolution pictures.
- Do you have a permission of reuse for figures 5 and 6?
- What specific aspects of schistosomiasis transmission should be prioritized in longitudinal studies to effectively assess climate change impacts over time?
- What does this scoping review add to the subject area compared with other published articles?
Author Response
Reviewer 3
Comment 1: The manuscript entitled "Impact of Climate Change on Schistosomiasis Transmission and Distribution - Scoping Review”,examines the impact of climate change on schistosomiasis transmission and its implications for disease control. Despite the good structure of the manuscript, there are some concerns which should be addressed.
Response: We thank the reviewer for the comments and constructive suggestions.
Comment 2: I think no need to write a structured abstract as this is scoping review not research article.
Response: We have modified the abstract to a non-structured abstract.
Comment 3: Pleas correct the word “Result” to be “Results”
Response: We have modified (see line 227)
Comment 4: All tables; please design as Three-line tables and make them fit to page borders.
Comment 5: Tables 1 and 3; can you reorder your content in an ascending years manner?
Response: We have modified the tables. Table 1 has been deleted and Table 3 has been modified and changed to Table 2.
.
Comment 6: Regarding figures; I think you need to provide high-resolution pictures.
Response: We have improved the image resolution.
Comment 7: Do you have a permission of reuse for figures 5 and 6?
Response: All the modified images are free to use and licensed under Creative Commons.
Comment 8: What specific aspects of schistosomiasis transmission should be prioritized in longitudinal studies to effectively assess climate change impacts over time?
Response: We have taken up this aspect and put the following into section…: ……. “Future research should prioritize longitudinal studies on the infectivity of snail intermediate hosts with Schistosoma and their capacity for sustained transmission to humans, to better understand the long-term impacts of climate change on schistosomiasis. Given the expanding geographic range of the disease, close monitoring of the presence or emergence of competent snail vectors in non-endemic areas is essential. This should involve active environmental surveillance for Schistosoma cercariae in water bodies, which serve as direct indicator of transmission risk. These ecological monitoring efforts must be complemented by public health surveillance in human populations, including parasitological diagnostics and, where appropriate, serological tools such as antibody screening to detect past or ongoing exposure. Integrated surveillance strategies are crucial for early detection of emerging transmission foci and for guiding timely, targeted interventions in at-risk regions. Additionally, expanding data collection to underrepresented areas is critical for building a more globally representative evidence base. Improved climate-health models that incorporate environmental, socioeconomic, and human-driven factors are urgently needed to enhance the accuracy of transmission predictions. Finally, empirical validation of predicted hotspots and studies on schistosomiasis co-infections are necessary to refine risk assessments and understand broader health impacts.”……………………..
Comment 9: What does this scoping review add to the subject area compared with other published articles?
Response: This scoping review adds significant value to the current body of literature by offering a comprehensive synthesis of how multiple climate variables, such as temperature, rainfall, and water body dynamics, jointly influence the ecology of snail vectors and the transmission of schistosomiasis. Unlike previous studies that often focus on single climatic factors or isolated regions, this review integrates diverse environmental drivers to present a more holistic understanding of disease risk under climate change. It is among the few reviews to identify emerging schistosomiasis hotspots in regions previously considered non-endemic and to highlight how climate change may shift or contract endemic zones, particularly emphasizing areas like East Africa, Southeast Asia, and parts of South America. Furthermore, the review distinguishes itself by systematically evaluating the role of climate adaptation policies, including water resource management, sanitation improvements, and early warning systems in mitigating transmission. It also fills a critical gap in linking climate science with public health strategies by advocating for the integration of climate adaptation into schistosomiasis control programs, especially in regions facing increased vulnerability due to climate variability.
Reviewer 4 Report
Comments and Suggestions for Authors
Dear Authors
The manuscript titled “Impact of Climate Change on Schistosomiasis Transmission and Distribution - Scoping Review” brings together the results of studies conducted in different parts of the world to interpret the effects of climate change on Schistosoma species. Considering that species in the Schistosoma genus threaten both human and animal health, it is evaluated that the information obtained within the scope of the study will contribute to the establishment of protection and control methods for pathogens in this genus. When the manuscript was examined from beginning to end, it was seen that the authors wrote the information in a way that the reader could easily understand.
Abstract
In the abstract section of the manuscript, the authors have given the importance of the study, the purpose of the study, the methods used in the study, and the results obtained as a result of the study in a fluent and easy-to-understand manner.
Introduction
In the introduction of the manuscript, the authors provide extensive information on the biology of Schistosoma, its clinical symptoms in humans, and the effects of global warming on the epidemiology and biology of the causative agent.
Line 55. Please write “Vertebrate Hosts” instead of “Human”. Because not only humans are important in the life cycle of Schistosoma, but also other vertebrate hosts are important in the biology of the causative agent.
Methodology
In this part of the manuscript, the authors provide detailed information about the selection criteria of the studies used in the study, the process of identifying the relevant publications, the determination of the information to be used in the selected publications, and the evaluation processes of this information.
Line 190. Please delete one of the two spaces after “A third reviewer”.
Results
In this chapter, the authors bring together the effects of climate change on Schistosoma infections obtained from different studies under various headings. It was observed that many tables and figures were used in this section.
In Table 1, it is suggested that the typefaces be revised and the same font and font size be used.
Slug genus/species names should be italicized in Table 1.
Line 331 Please replace Turkey with Türkiye.
Line 369. Multiple spaces before “mitigate” should be corrected.
Discussion
In the discussion section of the manuscript, the results obtained within the scope of the study are presented in a very explanatory and understandable manner.
References
It is recommended that the References section be thoroughly revised and that parasite names be italicized.
Author Response
Reviewer 4
Comment 1: The manuscript titled “Impact of Climate Change on Schistosomiasis Transmission and Distribution - Scoping Review” brings together the results of studies conducted in different parts of the world to interpret the effects of climate change on Schistosoma species. Considering that species in the Schistosoma genus threaten both human and animal health, it is evaluated that the information obtained within the scope of the study will contribute to the establishment of protection and control methods for pathogens in this genus. When the manuscript was examined from beginning to end, it was seen that the authors wrote the information in a way that the reader could easily understand.
Response: Thank you for the comment.
Abstract
Comment 2: In the abstract section of the manuscript, the authors have given the importance of the study, the purpose of the study, the methods used in the study, and the results obtained as a result of the study in a fluent and easy-to-understand manner.
Response: Thank you.
Introduction
Comment 3: In the introduction of the manuscript, the authors provide extensive information on the biology of Schistosoma, its clinical symptoms in humans, and the effects of global warming on the epidemiology and biology of the causative agent.
Response: Thank you.
Comment 4: Line 55. Please write “Vertebrate Hosts” instead of “Human”. Because not only humans are important in the life cycle of Schistosoma, but also other vertebrate hosts are important in the biology of the causative agent.
Response: We have modified, “human” has been changed to “vertebrate host”.
Methodology
Comment 5: In this part of the manuscript, the authors provide detailed information about the selection criteria of the studies used in the study, the process of identifying the relevant publications, the determination of the information to be used in the selected publications, and the evaluation processes of this information.
Response: Thank you.
Comment 6: Line 190. Please delete one of the two spaces after “A third reviewer”.
Response: We have deleted the space.
Results
Comment 7: In this chapter, the authors bring together the effects of climate change on Schistosoma infections obtained from different studies under various headings. It was observed that many tables and figures were used in this section.
Comment 8: In Table 1, it is suggested that the typefaces be revised and the same font and font size be used.
Comment 9: Slug genus/species names should be italicized in Table 1.
Response: We have modified the tables. Table 1 has been deleted from the manuscript and the Table 3 has been revised.
Comment 10: Line 331 Please replace Turkey with Türkiye.
Response: “Turkey” has been changed to “Türkiye”.
Comment 11: Line 369. Multiple spaces before “mitigate” should be corrected.
Response: We have removed the spaces.
Discussion
Comment 12: In the discussion section of the manuscript, the results obtained within the scope of the study are presented in a very explanatory and understandable manner.
Responses: Thank you.
References
Comment 13: It is recommended that the References section be thoroughly revised and that parasite names be italicized.
Response: We have italicized the scientific names in the references.
Reviewer 5 Report
Comments and Suggestions for Authors
Thoroughly prepared manuscript and relevant topic. Some comments:
a) Was it not the case of global COVID-19 lockdowns in 2020, 2021? There were not relevant reports on impact of climate change as a result of diminished human activity? Shouldn't that be a good example in your discussion to explore and discuss?
b) Not sure how much value Figure 5 adds. I mean we do know fluctuations occur in temperature and precipitation etc.... Seems to mostly occupy volume, than add real value without any predictive element.
c) Table 1 is long and can be tiring for the reader....Any way to group together similar patterns?
d) Methodology section too long (2.5 and 2.6) is basically repeating the obvious. Trim methods or group to make it more reader friendly the manuscript
e) Feels like AI-technology was employed in this manuscript. Kindly acknowledge if so...
Thank you
Author Response
Reviewer 5
Comment 1: Thoroughly prepared manuscript and relevant topic. Some comments:
Response: Thank you.
Comment 2: a) Was it not the case of global COVID-19 lockdowns in 2020, 2021? There were not relevant reports on impact of climate change as a result of diminished human activity? Shouldn't that be a good example in your discussion to explore and discuss?
Response: That is a very interesting dynamic that needs to be considered, however, climate change cannot be detected within 1-2 years. Thank you for the suggestion. We have includedthe following sentences into the discussion: “The COVID-19 lockdowns in 2020 and 2021 offer a compelling example of how reduced human activity can alter environmental dynamics and, potentially, disease transmission patterns. During these periods, countries worldwide reported significant improvements in water and air quality, reductions in industrial emissions, and changes in land use pressures. Indeed, CO2 emission was drastically reduced during the COVID-19 pandemic due to the diminished activity, but if this influences the climate change/rise of temperature globally cannot be evaluated. While the primary focus of global attention was understandably on pandemic containment, these unintended environmental effects highlighted the sensitivity of ecological systems to anthropogenic activity. Though limited empirical data exist specifically linking the lockdowns to shifts in schistosomiasis transmission, the broader implications are clear: reduced pollution, less water contamination, and slowed infrastructure development may have temporarily influenced/modified the creation of snail habitats which possible favours the snails survival and propagation. This period underscores the critical role human activity plays in shaping disease ecologies and reinforces the importance of sustainable, climate-smart policies. Integrating these lessons into climate adaptation planning could provide dual benefits for environmental restoration and infectious disease control, including neglected tropical diseases like schistosomiasis”. (Lines 493-510)
Comment 3: b) Not sure how much value Figure 5 adds. I mean we do know fluctuations occur in temperature and precipitation etc.... Seems to mostly occupy volume, than add real value without any predictive element.
Response: Thank you for the comment. Both rainfall and temperature affect the survival of the snails. So, we want to show the pattern of the fluctuations and the prevalence of schistosomiasis and treatment interventions. We observed that changes in rainfall and temperature correspond to changes in the schistosomiasis prevalence.
Comment 4: c) Table 1 is long and can be tiring for the reader....Any way to group together similar patterns?
Response: We have deleted table 1 from the manuscript and also modified table 3 to make it easier for the readers.
Comment 5: d) Methodology section too long (2.5 and 2.6) is basically repeating the obvious. Trim methods or group to make it more reader friendly the manuscript
Response: We have modified by combining sections 2.5 and 2.6 to: “Data extraction followed a standardized format to capture study and population characteristics, environmental factors (rainfall, temperature, water body changes), snail population dynamics, transmission trends, and the effectiveness of climate adaptation strategies. Priority was given to studies examining snail ecology and climate-driven shifts in schistosomiasis transmission. Extracted data were organized using spreadsheets or databases and analyzed through thematic and descriptive synthesis. This included mapping temporal and geographical changes in disease transmission, evaluating the ecological drivers affecting snail populations, and assessing the effectiveness of climate adaptation policies. The synthesis highlighted key findings, research gaps, and future directions, especially around predictive modeling and integrated control strategies.”
Comment 6: e) Feels like AI-technology was employed in this manuscript. Kindly acknowledge if so…
Response: Thanks for the comment. We did not employ any AI for the manuscript writing.
Round 2
Reviewer 5 Report
Comments and Suggestions for Authors
Thank you for addressing the comments! Congratulations!